# Umbilical Cord Blood Gas Parameters and Apgar Scoring in Assessment of New-Born Dogs Delivered by Cesarean Section

**DOI:** 10.3390/ani11030685

**Published:** 2021-03-04

**Authors:** Agnieszka Antończyk, Małgorzata Ochota, Wojciech Niżański

**Affiliations:** 1Wrocław University of Environmental and Life Sciences, Department and Clinic of Surgery, 50-366 Wrocław, Poland; agnieszka.antonczyk@upwr.edu.pl; 2Wrocław University of Environmental and Life Sciences, Department of Reproduction and Clinic of Farm Animals, 50-366 Wrocław, Poland; wojciech.nizanski@upwr.edu.pl

**Keywords:** dog, neonate, Apgar score, umbilical cord blood, Cesarean section

## Abstract

**Simple Summary:**

The article presents the results of the clinical evaluation and umbilical cord blood analysis obtained from newborn pups delivered by the elective Caesarean section. In human medicine both the umbilical cord blood and Apgar score were proved to provide valuable information on neonatal status. In veterinary medicine very few reports concerning the relation between the clinical neonatal assessment (Apgar score) and umbilical cord blood parameters exist. All puppies show mild respiratory acidemia regardless the Apgar scores result. The lactates were not elevated whereas all the puppies with low Apgar sores had higher glucose and mortality rates. Only pups with low initial Apgar scores were at risk of death within first 24 h. Adaptation to the extra-uterine life is crucial and any practical improvement in neonatal diagnostics and care would be beneficial for newborn puppy survival.

**Abstract:**

The article presents the results of the clinical evaluation (Apgar scores, AS) and umbilical cord blood gas analysis (UCBGA) obtained from clamped umbilical cords of newborn pups delivered by the elective Caesarean section. The study was planned as a controlled clinical study, the newborns were allocated into one of the groups, I—critical neonates (severe distress, AS ≤ 3), II—weak neonates (moderate distress, AS 4–6) and III—healthy neonates (no distress, AS ≥ 7). The following parameters were evaluated: pH (pH units), carbon dioxide partial pressure (pCO_2_; mmHg), oxygen partial pressure (pO_2_; mmHg), actual bicarbonate (cHCO_3_^−^; mmol/L), total carbon dioxide (cTCO_2_; mmol/L), base excess of extracellular fluid (BE(ecf); mmol/L), base excess of blood (BE(b); mmol/L), oxygen saturation (csO_2_; %), lactate (Lac; mg/dL), hematocrit (Hct; %PCV), hemoglobin (cHgb; g/dL), glucose (Glu; mg/dL), ions (Na, K, Ca, Cl). The majority of puppies had low AS at birth (AS 4–6 in 38.1% and AS ≤ 3 in 57.1% of the neonates), but most of them (85.7%) improved by the 20th min. reaching AS of 7 and more. Moreover, puppies with lower AS (≤ 3) were at higher risk of death within the first 24h (20.8% did not survive). The positive correlation was found between Apgar score measured at 0 min and pH (r = 0.46, *p* = 0.01), and between Apgar score (at 0 min) and base excess in whole blood measured [BE(b)] r = 0.36, *p* = 0.03). Whereas, a negative correlation was detected between Apgar score at 0 and 5th min and glycemia (r = −0.42, *p* = 0.05, r = −0.34, *p* = 0.02 respectively. Overall, the puppies with higher glucose levels had lower Apgar scores and were at higher risk of death. Furthermore, in our study, the newborn puppies had mild acidemia with elevated pCO_2_ levels and the HCO_3_ at the lower range of normal limits, suggesting the mixed component in the acidemic state. Adaptation to extra-uterine life is crucial and any practical improvement in neonatal diagnostics and care would be beneficial for newborn puppy survival.

## 1. Introduction

The transition from fetus to neonate is a crucial time of physiological adaptation. In fetal life, the oxygen is delivered through placental circulation, carbon dioxide and other fetal waste products go back to the mother’s circulation same route. In most cases, birth asphyxia and postnatal related problems originate from the interruption of uterine, placental or umbilical circulation. The transition from fetal to neonatal life is a crucial moment marked by profound changes to the physiology and biochemistry of all organs of the body. If improval fails, these changes would result in fatal consequences and most often death of a neonate [1,2]. The natural sequence of events in normal delivery has been mastered over the ages to heighten the chances of a newborn to survive. The medical approach, apart from bringing new challenges, also offers the opportunity to closely supervise and control the neonatal state at birth. Proper diagnosis and relevant clinical care would play an important role in increasing puppy survival [3]. Currently, in veterinary medicine, it is essential to better understand the pattern of changes which occur in a neonate at the moment of birth and its influence on the newborn homeostasis and performance [4,5].

The observed mortality rate in puppies is still high, around 20% [6,7,8,9]. Sadly, in veterinary medicine, weaker puppies often die underdiagnosed, due to clinical and therapeutic restrictions. The particularities of canine newborns, their size and physiology make it very difficult to develop or adopt standardized measures for neonatal diagnostics, assistance and direct therapy [10,11] contributing to inaccurate diagnosis and probably explaining the high mortality level [9]. The early and accurate detection of factors contributing to newborn puppies’ fatalities could improve neonatal outcomes [3,12]. 

The standard care in human neonatal assessment includes Apgar scoring (AS) and detailed clinical examination [13,14]. However, in neonatal intensive care units, the umbilical cord blood gases evaluation is also performed and considered essential [15]. Similar to human medicine for some years now the most advanced veterinary hospitals use modified Apgar scores to assess newborn puppies and kittens [16]. Similarly to human medicine, it quickly summarizes the newborn state and helps to decide what level of medical or emergency care is needed. In veterinary medicine, where usually more than one puppy or kitten is delivered, it has also another benefit—to help the staff to quickly determine the state of each puppy in the litter and set priorities which one needs urgent or secondary attention [3]. A few papers have been published on Apgar score use in veterinary medicine [3,9,12]. The systematic approach in newborn puppies evaluation proved to be effective in the timely detection of weak puppies with low Apgar scores that need urgent attention. It is very useful for short-term prognosis and also correlates with the future survival rate [8]. In addition to the current newborn status, the evaluation of umbilical cord blood gases analysis (UCBGA) proved to be an objective measure for the evaluation of fetal oxygenation status and metabolic acidosis at the moment of delivery [9,15]. The UCBGA has been shown to be more reliable than the Apgar scoring system in providing objective evidence of birth asphyxia and is now widely recommended especially for high-risk deliveries [17] in human medicine. The most important measurements are the pH, partial pressure of carbon dioxide (pCO_2_) and base deficit (BE), which help to determine whether the acidosis is metabolic or respiratory. Apart from these, other parameters can be assessed like ions, glucose, lactate or PCV, all providing valuable information on prenatal uterine condition, placental perfusion or impaired fetal nutrient supply [18]. Currently, not many publications are available concerning UCBGA in dogs, most probably due to different umbilical cord morphology in dogs than in humans and significantly lower birth weight and thus limited vessel size in puppies. Silva et al. studied only venous gases [9], whereas Mila et al. evaluated blood glucose, lactate and β-hydroxybutyrate concentration [7], and Groppetti et al. correlated neonatal umbilical cord blood lactate concentration with neonatal viability [12]. All the publications aimed at finding a relationship between studied parameters and postnatal puppy vitality and survival. Unfortunately, limited data, different time schedules, and the variety of vessels used to obtain the blood make it very difficult to compare the presented results. Since the veterinary neonatology is still lacking thorough evaluation of umbilical cord blood parameters and its correlation with neonatal health and performance, the main objective of this study was to describe the umbilical cord blood parameters indicating the character and level of biochemical and physical changes that occur at the time of the transition from fetal to neonatal state in newborn puppies delivered through Cesarean section.

## 2. Materials and Methods

### 2.1. Animal Selection

The 18 pregnant bitches of different breeds, including English Bulldog, German Shepherd, Boston Terrier, French Bulldog, Yorkshire Terrier, Welsh Corgi Pembroke, Great Dane, Shar-pei, Bull Terrier, Labrador, Shetland Sheepdog, were enrolled in the study. The mean bodyweight of the females was 28.11 kg (3.5–77 kg) and the mean age was 3.83 (1.5–7). The decision on elective Cesarean section was made by the vet responsible for the surgical procedure, based solely on clinical indications only and independently of this research. The patient qualification, time schedule and the veterinary procedures were performed under the Good Clinical Practice (GCP) guidelines. The Cesarean section was scheduled based on progesterone measurements during heat and LH peak determination and was performed on 63–64 days after LH peak, depending on the clinical evaluation of the dam, presence of milk and US foetal monitoring (heart rate). 

### 2.2. Anesthetic Protocol

All females underwent standard anesthetic protocol used at our Clinic for elective Cesarean sections. After admission the bitch was clinically assessed, weighted, and an intravenous catheter was placed, the preoxygenation was started and a single dose of 0.2 mg/kg s.c. meloxicam was administered (Metacam^®^ 5 mg/mL, Boehringer Ingelheim, Warsaw, Poland). Anesthesia was induced with intravenous bolus administration of propofol at 4–6 mg/kg to effect (Propofol-Lipuro^®^, 10 mg/mL B. Braun Melsungen AG, Germany), then the bitch was intubated and the anesthesia was maintained with isoflurane in oxygen (1.5–2.5%, IsoVet^®^, Piramal Healthcare, Morphet, United Kingdom ^®^). Intravenous fluids (crystalloids) were given in each case at the rate of 5 mL/kg/h. After removal of the last puppy opioids: morphine 0.3 mg/kg, (Morphini Sulfas^®^, WZF Polfa Warsaw, Poland), or buprenorphine 0.02 mg/kg (Bupaq Multidose^®^ Richter Pharma AG, Wels, Austria), or methadone (Comfortan^®^, Eurovet Animal Health BV, Bladel, Holand) 0.3 mg/kg were given to provide further analgesia [19].

### 2.3. Surgical Procedure and Umbilical Cord Sampling

A standard ventral midline approach was performed, the skin, subcutaneous tissue and abdominal wall were incised to enter the abdominal cavity and expose the uterine body and horns. Then, the edges of the abdominal incision were protected with moist swabs. Thereafter, the uterine incision was performed at a ventral surface of the uterine body and the closest neonate was delivered immediately. Directly after the puppy extraction from the uterus the fetal membranes were torn and removed from the muzzle and the umbilical cord was double clamped before placenta detachment. First, forceps were placed as close to the placenta as possible, and the second forceps about 1 cm from the newborn’s abdominal wall. The clamped vessel was then positioned between clamps to allow blood collection. At least 100 mcl of mixed umbilical cord blood was collected into a heparinized syringe with 25G needle, and taken for further analysis. The placenta and the remaining fetal membranes were removed using gentle traction. The same procedure was performed for all the fetuses, then the uterus was carefully checked for remaining fetal membranes, placentas or debris and sutured with a single continuous pattern (Monosyn^®^ 3/0, B. Braun Aesculap Chifa Sp. z o.o., Nowy Tomyśl, Poland). The abdominal cavity was lavaged with warmed saline before closing. The midline was sutured with a single continuous pattern and synthetic absorbable material, similarly, the subcutaneous tissue was sutured using absorbable material and continuous pattern, then the skin was routinely closed with absorbable material and intradermal pattern. 

### 2.4. Neonatal Care and Resuscitation

Depending on the size of the litter, 2–3 staff members with similar experience and skills gave first aid and, if necessary, performed resuscitation. Immediately after placental detachment, each puppy was placed on a heated blanket, the heart rate (HR), respiration rate (RR), mucous membrane colour, muscle tone and reflex irritability were briskly evaluated. Then, depending on the initial state, neonatal care or resuscitation was initiated. Neonatal care began with removing free fluid from the upper airways using the syringe bulb and clamping of the umbilical cord. Thereafter breathing was stimulated by a vigorous rubbing with a towel and the puppy was checked for obvious physical defects. The responding pups, i.e., vocalizing and breathing spontaneously, were moved into a veterinary incubator (Rcom MX-B90N, Autoelex Co., Ltd. Gyeongsangnam-do, Korea). The non-improving pups underwent neonatal resuscitation. If the heartbeat was present, the oxygen (constant flow or mask) was supplied and gentle stimulation of the abdominal area was started. The GV26 acupuncture point was stimulated and vigorous massage was continued. In puppies with asystole cardiac compression was performed at a rate of 120 per minute and epinephrine (0.01 mg/kg starting dose and continued if necessary at 0.1 mg/kg) was administered in the jugular vein. If there were no improvement after 20 min of intensive care it was stopped and the puppy was considered dead. All puppies were evaluated at 5th and 20th min using Apgar scoring and then a full clinical examination was performed around the moment of discharge to make sure all neonates can be reunited with the dam and sent home. Each owner was contacted by phone 48 h and 7 days after Cesarean section to confirm the survival/death of the puppies.

### 2.5. APGAR Scoring (AS)

The Apgar scoring was always done by the same person, using a modified Apgar scoring system adapted to be used in bitches [16]. The first evaluation was performed just after delivery, before first aid or neonatal care was instituted (0 min) and at the following times: 5th, 20th. The final evaluation was carried out around the time of discharging, i.e., 60th–90th min after birth. The AS reference range for the heart rate (HR) was: >220 bpm (2 points), 180–220 bpm (1 point) to <180 (0 points). Similarly, for the respiration rate (RR): >15 breaths per minute (2 points), 6–15 (1 point) to <6 (0 points). The reflex irritability detected after a gentle compression of a tip of a paw was evaluated depending on the degree of reaction: crying and quick leg retraction (2 points), weak leg retraction and no or just weak vocalization (1 point), and no leg retraction and no vocalization (0 points). Spontaneous movement of a newborn was rated as: 2—strong movement, 1—weak movement, and 0—absent. The color of mucous membranes was assessed as dark pink (2 points), light pink (1 point) and pale to cyanotic (0 points). All points together provided the final Apgar score: 7–10 no distress, healthy newborn; 4–6 moderate distress, weak newborn; and 0–3 severe distress, critical newborn [16].

### 2.6. Umbilical Cord Blood Gas Analysis (UCBGA) 

The blood samples for UCBGA were collected in heparinized syringes and analyzed immediately after sampling (POC: Point-Of-Care testing) using EPOC VET handheld analyzer (Siemens Healthineers, Erlangen, Germany). The following parameters were evaluated: pH (pH units), carbon dioxide partial pressure (pCO_2_; mmHg), oxygen partial pressure (pO_2_; mmHg), actual bicarbonate (cHCO_3_-; mmol/L), total carbon dioxide (cTCO_2_; mmol/L), base excess of extracellular fluid (BE(ecf); mmol/L), base excess of blood (BE(b); mmol/L), oxygen saturation (csO_2_; %), lactate (Lac; mg/dL), hematocrit (Hct; %PCV), hemoglobin (cHgb; g/dL), glucose (Glu; mg/dL), ions (Na, K, Ca, Cl). The blood gas analysis was normalized by puppies’ body temperature, estimated as 37 °C. 

### 2.7. Study Design

The study was planned as a controlled clinical study, the newborns were allocated into one of the groups, I—critical neonates (severe distress in AS evaluation), II—weak neonates (moderate distress in AS evaluation) and III—healthy neonates (no distress in AS evaluation), depending on the initial Apgar points obtained immediately after delivery (0 min). Neonates classified as critical received no more than 3 Apgar points in the initial examination, weak neonates received 4–6 points, whereas healthy received 7 and more Apgar points. Thereafter each puppy was re-examined using AS at 5th and 20th min. Finally, before discharging (around 60–90 min after delivery) each puppy was evaluated again. 

The initial (0 min) AS allowed to identify the weakest puppies which would need the most intensive care and to see whether this initial Apgar score would correlate with UCBGA. The following AS at 5 min was carried out to determine if the neonatal state improved after the neonatal care or resuscitation provided and to select puppies that need further attention and those which can be placed in the incubator. The following evaluation was performed at 20th min to check the status of all pups and to decide on stopping the resuscitation in non-responding critical neonates [1]. The last AS was carried out before discharging to make sure all neonates can be reunited with the dam and discharged home. Each owner was contacted by phone 48 h and 7 days after Cesarean section to confirm the survival/death of the puppies. 

### 2.8. Statistical Analysis

Puppies were divided based on the initial Apgar score results into groups of I—critical neonates, II—weak neonates and III—healthy neonates. Statistical analysis included standard descriptive statistics and normality testing using the Kolmogorov–Smirnov test with the Lilliefors correction. One way repeated measures analysis of variance was used to compare Apgar scores within different time points. The results of UCBGA were compared among groups using one-way analysis of variance. In the case where the third group was too small, a *t*-test was used to compare two sets of data. The correlation between UCBGA and the Apgar scores results were evaluated with Spearman’s correlation test. The significance level was set to 0.05 in all the abovementioned tests. Calculations were performed using Statistica 13 (StatSoft, Kraków, Poland).

## 3. Results

Eighty-five puppies were born in 18 Cesarean sections. One to six puppies were delivered in each Cesarean section. The collection of umbilical cord blood was successful in 42 of them (1 to 5 puppies per dam), and these pups were enrolled in the study, however, in the remaining 43, the amount of the collected blood was not enough to perform analysis. 

Statistical analysis revealed differences between Apgar scores obtained at different time points (*p* < 0.00001) (Table 1). At 0 min of birth, only 4.8% of the neonates (n = 2) had an Apgar score of 7 or above, which was the cut-off value for considering the puppy as normal. In total, 38.1% (n = 16) and 57.1% (n = 24) of the neonates had Apgar scores of 4–6 and of 3 or below, respectively. At 5 min of birth, 31.0% (n = 13) pups were considered normal with an Apgar score of 7 or above. Furthermore, 52.4% (n = 22) and 16.7% (n = 7) had Apgar score of 4–6 and 3 or below, which classified the pup as weak or critical, respectively. At 20 min of birth, the vast majority of newborns (85.7%, n = 36) were scored at 7 or higher. Four newborns (9.5%) were classified as weak (4–6 AS), however, critical pups were also found (4.8%, n = 2). Two newborns did not survive and had an Apgar score of 0 at 20 min.

The initial Apgar score (AS) assessment was indicative of further neonatal performance. All puppies classified with Apgar score ≥7 were in good condition in the following assessments, receiving similar or higher Apgar scores. Similarly, in the group of puppies classified as weak (AS 4–6 at 0 min), all improved enough to be classified as normal (AS ≥ 7) at 20th min despite the fact that some of them 43.75 % (n = 7) still remained weak (AS 4–6) at 5th min. However, in the group of critical neonates at 0 min, only 8.3% (n = 2) improved enough to receive AS ≥ 7 in the 5th min, but in the 20th min, as many as 75 % (n = 18) improved significantly and were classified with AS ≥ 7. The remaining 58.33 % (n = 14) received only AS of 4–6 at 5th min, however, only 16.67 % (n = 4) of them still remained weak and received AS 4–6 at 20th min. Unfortunately, the 33.33 % (n = 8) of pups were still critical at 5th min and two (8.33%) of them died before the next assessment in 20th min (Figure 1), while three out of the four puppies assessed as weak at 20 min died within the first 24 h (12.5%). The remaining 37 pups were healthy and were developing normally in the final check on the 7th day after birth.

The mean results of umbilical cord blood gas analysis including all the investigated parameters are presented in Table 2. The average pH was 7.2, which in adult dogs indicates mild acidemia, however, there is no reference range for newborn puppies. Additionally, all the investigated puppies had higher mean pCO_2_ comparing to reference values in adults (35–45 mmHg). The mean glucose level was also quite high (72.64 mg/dL) and correlated negatively with the AS evaluation, while the mean lactates were within the normal adult dog range (2.45 mmol/L) (Table 2). 

The retrospective analysis of umbilical cord blood gas analysis (UCBGA) parameters in puppies classified as critical, weak, or healthy at different time points of the study (0, 5th, 20th min) was also performed (Table 3). Spearman’s correlation test showed the presence of a positive correlation between Apgar score measured at 0 min and pH (r = 0.46, *p* = 0.01) and between Apgar score (at 0 min) and base excess in whole blood measured [BE(b)] r = 0.36, *p* = 0.03). Whereas, a negative correlation was detected between Apgar score at 0 min and glycemia (r = −0.42, *p* = 0.05) as well as Apgar score measured at 5 min of birth (r = −0.34, *p* = 0.02). No statistically significant correlation was found between any of the other blood gas parameters examined.

## 4. Discussion 

Identification of newborn puppies at high risk of developing adverse outcomes after delivery is the main goal in advancing veterinary neonatology. The presented research intended to give a chance for a better understanding of basic chemical and physical changes which occur in a neonate just after birth and to establish the most apparent clinical and biochemical findings influencing the early neonatal life. In human medicine, both the umbilical cord blood and Apgar score were proved to provide valuable information on the neonatal status. To the best of the authors’ knowledge, this is the first report presenting the full umbilical cord blood gas analysis in newborn puppies together with the clinical evaluation of their health state.

In dogs, same as in human medicine, the initial newborn puppy assessment should include Apgar scoring, however, based on the obtained results and the available literature, the Apgar score obtained just after birth (0 min) seems to be less predictive for further puppy outcome in comparison to the next evaluation carried out few minutes after birth. It has been noted in human babies that the transition time may vary in normal infants, so the low initial Apgar score is often due to a transient low oxygen saturation, which resolves spontaneously when normal respiration is established [20]. Hence, in humans, the Apgar score is routinely measured at 1 min and at 5 min after birth [21]. In this study, most of the newborns who scored as critical (n = 24) immediately after birth, improved and developed satisfactory condition in the following evaluations (75% of puppies had AS ≥ 7 in 20th min after birth). Our Apgar score results at 0 min were much lower than those reported by Vilar et al. [22], however, it might be due to the different surgical and anesthetic protocols used or the exact time of the initial Apgar assessment, which in our study was performed prior to any medical attention provided for a newborn. The exact time of cord clamping could also influence the neonatal state as reported by Pereira et al. who described a higher vitality in puppies in which the umbilical cord was preserved intact for a least 3 min after birth to allow the residual blood to flow to the puppies [23]. On the other hand, Davidson [1] suggested that neonates delivered via Cesarean section tend not to initiate respiration spontaneously, which may affect their initial Apgar score records. Considering the above, the vitality scoring performed immediately after birth seems to be useful in predicting short-term abnormalities, but it does not provide information on the long-term outcome, and the way of puppy handling might play a role in the subsequent neonatal performance.

In dogs, some authors measured the Apgar score for the first time at 5 min after birth, which same as in our investigations, corresponded much better with the further puppy performance [3,9,22,24,25]. Nevertheless, the authors of this paper agree with Batista et al. [3] that the initial (0 min) Apgar assessment should still be performed, mostly to help the veterinary staff in triage and evaluating the amount of care a pup would need. Contrary to human medicine, where typically only one infant is born, veterinary personnel usually deals with several puppies. Therefore, early and accurate identification of the neonate’s condition is essential to define which newborn requires immediate assistance and to provide an adequate level of clinical care necessary to improve the viability and survival of each puppy [3]. The suitable first aid provided for a newborn might be crucial for its further performance and survival. It is worth noting that 11.9% (n = 5) of pups did not survive the first 24 h, despite neonatal critical care provided, indicating some predictive value of the initial Apgar scoring. 

The detailed comparison of Apgar score results obtained here and reported in the literature is difficult due to different time schedules of Apgar evaluation, mode of delivery and the potential influence of medications given to the dams. Some authors measured Apgar score only once [3,12,16], whereas others twice at 0 and 60th min [22], or three times at 0, 5th and 60th min [9] or at 5th, 15th and 60th min after delivery [24]. The adopted time schedule of Apgar score recording in our study aimed firstly (AS at 0 min) at recognizing the weakest newborns and indicating whether a puppy requires medical intervention. The second examination (AS at 5 min) helped to determine whether a neonate improved after the neonatal care or resuscitation provided. The following scoring (AS at 20 min) confirmed the status of neonates and allowed the decision to stop resuscitation in non-responding pups. All newborns underwent a full clinical examination before discharging to ensure all puppies are well enough to go home. In our clinical settings such time schedule turned out to be the most efficient for the veterinary staff involved in neonatal care and decision making process.

In human babies, the pH and BE possess the high prognostic value of an adverse neonatal outcome as they reflect respiratory status and acid–base imbalances [15,26]. When linked to the current newborn status it would help to determine the underlying cause of the blood gas changes. The same as other authors [25,27], we found the newborns to be mildly acidemic with a mean pH value of 7.2, while Lucio et al. [27] reported for vaginally born puppies the pH at 7.1 and Vassalo et al. [25], similar to our results, at 7.2 for both eutocia and Cesarean section. The acid production in newborns is buffered to maintain the pH within limits. However, if the delivery of oxygenated blood through the placenta is disrupted, hypoxia occurs and fetal metabolism shifts to the anaerobic pathway with the production of lactates. With high lactic acid production, the buffer system becomes insufficient leading to neonatal acidosis, low pH and poor Apgar scores [28,29]. We observed lower pH values in puppies classified as critical at birth (7.18 vs. 7.27 in puppies receiving 7 or more points in Apgar scoring at birth). The lower pH was positively correlated with BE increase and a trend towards higher pCO_2_ levels. However, the puppies examined in our research had low mean concentrations of lactates (2.45 mmol/L), bicarbonates on the lower border (22.78 mmol/L), decreased BE(b) (−6.04 mmol/L) and moderate increased partial pressure of CO_2_ (58.59 mmHg) which suggest both, respiratory and metabolic component as the reason of the pH changes. This is in agreement with other papers where authors reported mixed acidosis being the cause of poor neonatal performance [25,27]. Moreover, in our study puppies receiving high Apgar scores at birth (AS ≥ 7) had also lower pCO_2_ and higher pH compared to the rest of the investigated pups. Unfortunately, we had only two such puppies, which did not allow for the statistical analysis, but their results still suggest the respiratory component in the observed neonatal acidemia. Same as mentioned before, the discrepancies between our results and those reported by other authors may be due to different vessels used for obtaining the blood sample. We collected the mixed (venous–arterial) umbilical cord blood, whereas Lucio et al. and Vassalo et al. [25,27] collected venous blood from the jugular vein to evaluate neonatal acidemia.

Recently, several studies have examined the efficiency of the combination of Apgar scoring system with umbilical cord blood gas assessment (UCBGA), lactate measurement or acid–base analysis, proving to be a useful diagnostic tool especially in the intensive care units [15,26]. Articles published in human medicine showed that the measurement of lactate concentration in umbilical blood is a good predictive factor for neonatal outcome [30,31,32,33]. Further, the relationship between the type of delivery and the lactate content was reported, with the highest lactate level in babies delivered instrumentally and the lowest after Cesarean section [34]. Our results demonstrated the low mean value of umbilical lactate concentration (2.45 mmol/L). This result is consistent with the study of Groppetti et al. [12] who showed the lowest lactate levels in puppies born by elective Cesarean section (3.9 mmol/L) compared to the emergency procedures (6.3 mmol/L) or vaginal parturition (6.8 mmol/L). While, Castagnetti et al. [35] reported for vaginal delivery the initial lactate at the level of 6.7 mmol/L in surviving newborns, and 10 mmol/L in those that died within the first day of life. Hyperlactatemia would develop in the late stage of fetal hypoxia, when the lack of oxygen leads to anaerobic metabolism and the production of lactic acid [33]. Thus the mode of delivery and the uterine activity during labor, may contribute to the switching into the anaerobic condition and therefore affect the level of lactates. Moreover, Groppetti et al. [12] sampled the umbilical cord up to 5 min after delivery, whereas Castagnetti et al. [35] obtained the blood sample by puncture of puppy’s finger pad. The time of sampling in the first report and another vessel used for blood collection may explain the differences in the presented results. Groppetti et al. [12] also found a relationship between lactate concentrations in healthy and unhealthy neonates, and proposed the cut-off value for lactate at 5 mmol/l, to distinguish between healthy and distressed pups. However, we did not observe such a correlation between lactate levels in newborns classified as critical and weak. Unfortunately, we were not able to include into this comparison the results obtained in pups classified with Apgar score over 7 at birth, due to the low number of cases (only two puppies), however their lactate levels were 2.24 and 3.38 mmol/L respectively. Interestingly, the two critical puppies which died within first 20 min after delivery had higher initial lactate levels (4.42 and 3.3 mmol/L, respectively).

We also noted differences in the glucose levels between puppies classified in different time points (0 and 5 min) as critical (AS ≤ 3), weak (AS 4–6) and normal (AS ≥ 7). Puppies classified based on Apgar score as still being critical at 5 min, had significantly higher levels of glucose in comparison to the weak and normal neonates. Critical pups from 0 and 5th min showed also a negative correlation between AS and glucose levels, which may indicate its impact on the newborn clinical state. The available literature does not provide sufficient information on the relationship between glucose concentration in umbilical blood and neonatal survival in dogs. Castagnetti et al. [35] observed a tendency to higher glucose levels in non-surviving comparing to surviving neonates, which is in agreement with our results. On the contrary, other studies showed that the low glucose levels were related to poor newborn puppy performance. Vessalo et al. [25] found that newborn puppies with glucose levels below 40 mg/dL had usually low Apgar scores and poor reflexes (suckling, rooting and righting reflex). Our results showed the opposite relationship: pups receiving 7 and more Apgar points in the initial evaluation had lower mean glucose concentration (30.5 ± 3.53 mg/dL). In humans, similarly to our results, Khan et al. [36] showed that babies with fetal distress (identified by the meconium-stained fetal fluid and fetal bradycardia) had higher glucose levels in the umbilical cord blood. Since, the glucose concentration decreases with the impaired utero-placental perfusion or increased fetal consumption [18], the mode of delivery in our study (elective Cesarean section) vs. eutocia and dystocia resolved via Cesarean section in Vassalo et al. [25] research may explain the presented discrepancies. These findings were also confirmed by Lucio et al. (2021) showing differences in maternal and neonatal glucose levels depending on delivery mode [37]. Authors found hyperglycemia in puppies from C-section and fetal dystocia groups whereas puppies from eutocia and maternal dystocia remained normoglycemic. One of the proposed theories suggested hyperglycemia is the response to peripartum stress (increasing catecholamine, hypoxia, etc.). This explanation may support our findings that newborns with low Apgar scores had higher glucose. On the other hand, Lucio et al. [37] and Vanspranghels et.al [30] suggested that glucose level reflects rather maternal physiology than fetal status and did not appear to be useful for predicting neonatal morbidity. Moreover, the abovementioned concern regarding the vessel used for blood sampling could have also contributed to the differences in the presented results. Considering the above, further studies are necessary to explain these changes in the reported results.

Although puppies born via elective Cesarean section may initially present asphyxia and obtain low Apgar scores, in most of the cases, their state improves substantially within the next few minutes when neonatal care is provided and the proper respiration is established. In the light of the presented results, the umbilical cord blood collection was feasible, and the cord vessels provided sufficient blood for further analysis. UCBG analysis contributed to obtaining new and important data regarding newborn metabolic states at the transition time. If further investigated, the umbilical cord blood may allow us to set the reference values for the evaluated parameters and present association with medium and long-term neonatal outcomes. The improvement of neonatal assessment protocol should elucidate all factors and circumstances affecting the neonatal state, provide practical knowledge for practicing veterinarians, and hopefully decrease the mortality rates in puppies.

## 5. Conclusions

The detailed evaluation of a newborn state is crucial in the overall understanding of neonatal physiology. In our study, the newborn puppies had mild acidemia with elevated pCO_2_ levels and the HCO_3_ at the lower range of normal limits suggesting the mixed component in the acidemic state. However, the lack of reference values in the available literature makes it very difficult to interpret the obtained data. Contrary to what was expected, the lower Apgar score results were, the higher glucose levels were found in newborns, and higher mortality rates were observed. Only pups with the lowest initial Apgar scores were at risk of death within the first 24 h. 

It is necessary to increase the knowledge and understanding of clinical and metabolic changes in newborn puppies, since the puppy mortality rate is still high in veterinary medicine. Adaptation to the extra-uterine life is crucial and any practical improvement in neonatal diagnostics and care would be beneficial for newborn puppy survival. 

## Figures and Tables

**Figure 1 animals-11-00685-f001:**
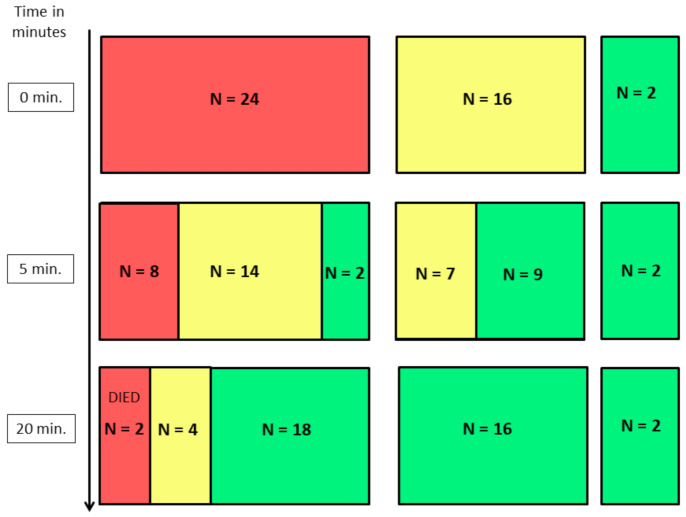
The following results of Apgar score assessments in puppies classified as critical (AS ≤ 3), weak (AS 4–6) and normal (AS ≥ 7) immediately after birth (0 min). red rectangles—newborns with Apgar score < 4; yellow rectangles—newborns with Apgar score 4–6; green rectangles—newborns with Apgar score > 6.

**Table 1 animals-11-00685-t001:** Summary of Apgar Score (AS) at 0, 5th and 20th min of birth (n = 42).

	AS 0 Min	AS 5th Min	AS 20th Min
Mean ± sd	2.7 ± 2 *	5.3 ± 2.3 *	7.9 ± 2.3 *
Min–max	0–8	0–9	0–10

* indicates significant difference between time points (*p* < 0.00001).

**Table 2 animals-11-00685-t002:** Mean values of umbilical cord blood gas parameters measured at birth (0 min, n = 42), mean ± SD.

	pH Units	pCO_2_ mm Hg	pO_2_ mm Hg	HCO_3_^−^ mmol/L	Na^+^ mmol/L	K^+^ mmol/L	Ca^++^ mmol/L	Cl^−^ mmol/L	Glu mg/dL	Lac mmol/L	Hct %	cHgb g/dL	BE(b) mmol/L	BE(ecf) mmol/L
mean SD	7.2 ± 0.1	58.6 ± 12.7	19.7 ± 13	22.8 ± 3.4	132.7 ± 6.1	3.5 ± 0.8	0.6 ± 0.4	109.5 ± 2.8	72.6 ± 24.7	2.5 ± 1.5	39.3 ± 11.5	13.6 ± 3.6	−6.0 ± 3.6	−5.2 ± 3.8
min	6.9	32.1	5	16.2	117.	2.1	0.3	105	18	0.9	10	3.9	−16.6	−14.6
max	7.3	84.5	69.3	29.7	144.0	5.8	1.7	119	131	9.3	62	21	−1.1	1

**Table 3 animals-11-00685-t003:** The comparison between the chosen parameters of umbilical cord blood gas analysis (UCBGA) collected at birth (0 min) in puppies classified with Apgar score (AS) as critical, weak or healthy at different time points of the study (0, 5th, 20th min).

Apgar Score	Parameters	0 Min	5th Min	20th Min
CRITICAL AS < 4		n = 24	n = 7	n = 2
	pH units	7.2 ± 0.1 *	7.2 ± 0.1	7.1 ± 0.01 ^A^
	pCO_2_ mmHg	60.2 ± 12.4	60 ± 13.2	69.7 ± 14.7
	BE(b) mmol/L	−7.1 ± 4 *	−7.8 ± 3.3	−9.2 ± 2.9 ^A^
	BE(ecf) mmol/L	−6.2 ± 4.2 *	−6.5 ± 3.4	−7.1 ± 4.5 ^A^
	Lac mmol/L	2.6 ± 1.7	2.3 ± 1.2	3.9 ± 0.8 ^A^
	Glu mg/dL	79.3 ± 21.8	95.3 ± 2.8 *	90 ± 33.9 ^A^
WEAK AS 4–6		n = 16	n = 22	n = 4
	pH units	7.2 ± 0.1 *	7.2 ± 0.1	7.2 ± 0.1
	pCO_2_ mmHg	57.6 ± 13.2	59.8 ± 13	54 ± 9.2
	BE(b) mmol/L	−4.5 ± 2.3 *	−5.9 ± 4.1	−7.2 ± 4.8
	BE(ecf) mmol/L	−3.7 ± 3 *	−5 ± 1	−6.8 ± 4.8
	Lac mmol/L	2.1 ± 1	2.4 ± 1	2.1 ± 0.1
	Glu mg/dL	67.9 ± 24.4	72.4 ± 15.3 *	72 ± 20
NORMAL AS > 6		n = 2	n = 13	n = 36
	pH units	7.3 ± 0.1 ^A^	7.2 ± 0.1	7.2 ± 0.1
	pCO_2_ mmHg	47.3 ± 10.1 ^A^	56.5 ± 13	58.5 ± 12.8
	BE(b) mmol/L	−5.6 ± 0.1 ^A^	5.3 ± 3	5.8 ± 3.4
	BE(ecf) mmol/L	−5.6 ± 0.6 ^A^	−4.7 ± 3.8	−4.9 ± 3.8
	Lac mmol/L	2.8 ± 0.8 ^A^	2.7 ± 2.1	2.4 ± 1.5
	Glu mg/dL	30.5 ± 3.5 ^A^	64.4 ± 29.2 *	71.8 ± 25

* Indicates significant differences between values in a column (*p* < 0.05); ^A^ Indicates values excluded from statistical analysis due to the low number of cases (n = 2).

## Data Availability

The data that support the findings of this study are available from the corresponding author [MO], upon reasonable request.

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
