# Peer review of "Umbilical Cord Blood Gas Parameters and Apgar Scoring in Assessment of New-Born Dogs Delivered by Cesarean Section"

_animals, 2021, doi:10.3390/ani11030685_

Round 1
Reviewer 1 Report
Original study with a new database about canine neonates and very well evaluated Apgar score with an evolution in the time. Complete bibliography and interesting comparison with studies performed in human neonatology.
Some minor errors in the presentation form and text editing (see attachment).
I accept this article after minor revision.

Author Response
The Authors thank the Reviewer for the review and comments which definitely helped us to improve our MS. Please find below the point-by-point answers to the Reviewer’s comments:
L3 - corrected
L16 - double space has been removed
L 24 - we have not removed the extra space between lines - it was created as a result of text formatting in the editorial office and we are not able to change it.
L31 - we have added the info about AS with a higher risk of death, as follows:
Moreover, puppies with lower AS (below 4) were at higher risk of death within first 24h (20.8% did not survive).
L48-50 - the sentence has been changed according to the Reviewer’s suggestions:
The transition from fetal to neonatal life is a crucial moment marked by profound changes to physiology and biochemistry of all organs of the body.
L108-110 - we have standardized the font size
L111 - The following reference was added: Guidelines on good clinical practices. The European Agency of the Evaluation of Medicinal Products, Veterinary Medicine and Information Unit; CVMP/VICH/595/1998, London, 04.07.2000; https://www.ema.europa.eu/en/vich-gl9-good-clinical-practices
L123 - we have changed the parentheses
L136 - has been changed
L151 - has been changed
L165 - dot has been removed
L169 - dot has been removed
L191 - double space has been removed
L208 - dot has been removed
L211 - double space has been removed
Table 2 - double spaces have been removed in the whole table
Table 3 - double space has been removed
L248 - double space has been removed
Reviewer 2 Report
I have only very minor changes which could improve the manuscript.
Animal selection:
Please add the breed of dogs included into the study
Please indicate needle size as for collecting the blood from umilical cord blood; how many ml did you collect?
M@M:
Please add BCS of dogs included into the study
Did you determine P4 before cesarian section?
I think that the main GCP procedures concerning s.c. should be added in points
Conclusion
In my opinion the first two sentences of that section should be transfered to Discussion section. It is the conclusion - so - only summary of the study should be written here
Author Response
We thank the Reviewer for appreciation of our work, valuable comments and suggestions.
Please find the detailed answers to the Reviewer comments below:
Animal selection:
Please add the breed of dogs included into the study
We have added the info about the dogs' breed enrolled in this study.
The 18 pregnant bitches of different breeds, including English Bulldog, German Shepherd, Boston Terrier, French Bulldog, Yorkshire Terrier, Welsh Corgi Pembroke, Great Dane, Shar-pei, Bull Terrier, Labrador, Shetland Sheepdog, were enrolled into the study
Please indicate needle size as for collecting the blood from umbilical cord blood; how many ml did you collect?
We thank the Reviewer for this suggestions. We have added this info to the Surgical Procedure and Umbilical Cord Sampling section, as follows:
At least 100 mcl of mixed umbilical cord blood was collected into a heparinized syringe with 25G needle and taken for further analysis.
M@M:
Please add BCS of dogs included into the study
We tried to include the BCS into the clinical examination, but all females enrolled into the study were highly pregnant, so it was difficult to obtain the objective and reliable measurements. It was decided not to include and analyse BCS in this study.
Did you determine P4 before cesarian section?
We did not determine P4 before the surgery. The P4 was evaluated during heat for LH peak determination. The decision on the elective Caesarean Section was made based on P4 measurements obtained during oestrus and C. section was performed on 63 - 64 day after LH peak, depending on the clinical evaluation of the dam, presence of milk and US foetal monitoring (mainly HR).
I think that the main GCP procedures concerning s.c. should be added in points
The procedure of Caesarean section was improved according to the reviewer suggestions, and the section Surgical Procedure and Umbilical Cord Sampling has been modified, as follows:
A standard ventral midline approach was performed, the skin, subcutaneous tissue and abdominal wall were incised to enter the abdominal cavity and expose uterine body and horns. Then, the edges of the abdominal incision were protected with moist swabs. Thereafter, the uterine incision was performed at a ventral surface of the uterine body and the closest neonate was delivered immediately. Directly after the puppy extraction from the uterus the fetal membranes were torn and removed from the muzzle and the umbilical cord was double clamped before placenta detachment. First forceps was placed as close to the placenta as possible, and the second forceps about 1 cm from newborn's abdominal wall. The vessels were gently stretched between forceps to allow the blood sampling. At least 100 mcl of mixed umbilical cord blood was collected into a heparinized syringe with 25G needle and taken for further analysis. Placenta and the remaining fetal membranes were removed using gentle traction. The same procedure was performed for all the fetuses, then the uterus was carefully checked for remaining fetal membranes, placentas or debris and sutured with a single continuous pattern (Monosyn® 3/0, B. Braun Aesculap Chifa Sp. z o.o., Nowy TomyÅ›l, Poland). The abdominal cavity was lavaged with warmed saline before closing. The midline was sutured with a single continuous pattern and synthetic absorbable material, similarly the subcutaneous tissue was sutured using absorbable material and continuous pattern, then the skin was routinely closed with absorbable material and intradermal pattern.
Conclusion
In my opinion the first two sentences of that section should be transferred to Discussion section. It is the conclusion - so - only summary of the study should be written here
The sentences have been moved to the beginning of the Discussion section according to reviewer suggestions.
Reviewer 3 Report
The present manuscript aims to propose a predictive analysis of neonatal health during the fetal-to-neonatal transition in surgically born puppies, based on umbilical cord blood acid-base balance. For this purpose, neonates were classified according to their vitality score at birth and prospective correlations with blood acid-base variables were evaluated.
This manuscript is interesting and, in fact, this is an important research area and studies on canine neonatology have to be stimulated for an overall understanding of neonatal physiology. The study on puppies’ assessment as a clinical parameter in neonatology should be encouraged. Despite the interesting area of experimentation, there are important issues that derail its publication in the present form. The material and methods section lacks some essential information.
My most important concern relates to the umbilical blood itself. While hemodynamically active, umbilical vessels blood content is of both maternal and fetal origin, unless you have previously isolated the umbilical vein. Therefore, you cannot affirm that the acid-base analysis you are performing is restricted to the neonate during the transition phase. You are possibly analyzing fetal blood mixed with maternal blood under the intra-uterine environment. Thus, the proposing analysis for the main goal of predicting short-term neonatal outcome is biased.
You should overall revise the manuscript for its syntax and punctuation, as the text is sometimes difficult to follow.
Some of the major comments and criticisms include:
Title: authors should refer to the birth mode and explicitly state that the manuscript refers to c-section born puppies.
Abstract
Line 21: Did you collect blood before or after umbilical cord clamping? Please, include this important information.
Line 23: please, include the cutoff Apgar value for each neonatal classification.
Line 36-39: this is not the conclusion of your work. Please, rewrite.
Introduction:
Line79-81: it is important to explain that human umbilical cord morphology is distinct from dogs. Moreover, the umbilical cord is used as an access cannulation site in babies, even after cord clamping. In other words, the blood collected is surely of neonatal origin at this time-point.
Line 86: In order to determine if the acidosis is respiratory, you should determine the pCO2. Please, clarify.
Material and methods:
You should place the full description of the study population in the MM section. Some very important information should be provided:
- How many puppies in total and how many puppies per bitch?
- It is important to mention how the vet estimated the date for performing the c-section. Were all pregnancies full-term?
- Please describe in detail the neonatal management: did the authors subject puppies to any stimulation procedure after birth?
- Did the authors perform a neonatal resuscitation maneuver before blood collection?
- Please, describe in detail how you performed the blood collection: by venous puncture (which needle caliper?) or by manual ejection.
- Surprisingly, authors have managed to collect umbilical blood after cord clamping, as the vessels walls collapse down immediately after blood flow interruption. Please, explicit all the difficulties in collecting umbilical blood.
- What was the volume of blood collected?
- Another important information is what vessel direction the umbilical blood was collected? The umbilical cord has a two-way direction of blood flow, i.e., from maternal or fetal origin. Please clarify if the blood was actually from the fetus / neonate.
- Please, detail what procedures you performed for neonatal resuscitation: manual ventilation, mechanical ventilation, exogenous oxygen, etc...
Line 110: I wonder how was possible to collect blood from the umbilical cord of a 3.5kg bitch.
Line 149-156: Was the blood acid-base analysis normalized by the neonate body temperature?
Line 168: please describe in detail the neonatal intensive care.
Line 182: revise your statistical method. The T Test is commonly used to determine if the means of two sets of data are significantly different from each other, which is not the case. You have three sets of data (or groups). Please, clarify.
Results:
Line 198-199: Have you examined puppies for any mal-formation or necroscopically examined post-mortem?
Lines 231-232: I do not think that comparing between the blood acid-base imbalance at birth and further AS (5 min and 20 min) is meaningful. It is better to perform a logistic regression analysis, calculating the odds ratio of each blood gas variable for predicting the first hour clinical outcome of the neonates.
Discussion
Line 260: On the other hand, we cannot predict future outcome, such as linear growth development, future behavioral changes, and social interrelationship. In other words, vitality score performed immediately after birth cannot predict long-term outcome. However, it may prove fruitful to predict short-term abnormalities.
Lines 261-262: Indeed, your vitality scores at birth are comparatively lower compared to previous studies (please, refer to Pereira et al., J Vet Med Sci 2020; 82(2):247-253), with special regards to the high frequency of severe distressed puppies.
Lines 262-263: If you performed the initial scoring before the immediate medical attention, probably it will be very low, almost resembling the intra-uterine environment. This has to be clearly stated in your methods and additionally discussed here.
Lines 275-276: even if you have performed an intensive critical care? Please clarify if the percentage of death includes the puppies in which you applied the reanimation protocol.
Line 268: what was your neonatal care protocol?
Line 287: what was your resuscitation protocol?
Line 288: The Apgar scoring was initially proposed (Virginia Apgar, 1952) to be performed around the first hour of life. Therefore, any clinical assessment of the newborn after 60 min should not be named Apgar score. Please, revise.
Line 321: this affirmation can only be made with the analysis of blood BE and bicarbonate, which are markers of the metabolic component of acidosis.
Line 326: important to say that this was vaginally born puppies, whereas your results relate to surgical birth.
Line 337: you cannot discharge an ongoing metabolic acidosis only by analysis lactate concentration. An overall interpretation of BE and bicarbonate should be done.
Lines 353-355: Despite not having an umbilical blood, please see Lucio et al. Domest Anim Endocrinol. 2021;74:106483.
Conclusion
Line 390-392: this is not a conclusion of your work.
Lines 392-393: you cannot conclude this.
Author Response
Authors would like to thank the Reviewer for comments and suggestions which helped to improve the MS. All the suggested changes have been addressed as follows:
Title: authors should refer to the birth mode and explicitly state that the manuscript refers to c-section born puppies.
The mode of delivery was added: Umbilical Cord Blood Gas Parameters and Apgar Scoring In Assessment of New-Born Dogs delivered by cesarean section
Abstract
Line 21: Did you collect blood before or after umbilical cord clamping? Please, include this important information.
The following information was added: obtained from clamped umbilical cord of newborn pups (Line 22)
Line 23: please, include the cutoff Apgar value for each neonatal classification.
The cut-off vaues were added, Lines 24 - 26
Line 36-39: this is not the conclusion of your work. Please, rewrite.
The sentence was changed (Lines: 37 – 42): In our study the newborn puppies had mild acidemia with elevated pCO2 levels and the HCO3 at the lower range of normal limits suggesting the mixed component in the acidemic state. Con-trary to what could be expected, the lower Apgar score results were the higher the glucose levels were found in newborns and the higher mortality rates. Only pups with the lowest initial Apgar scores were at risk of death within first 24h.
Introduction:
Line79-81: it is important to explain that human umbilical cord morphology is distinct from dogs. Moreover, the umbilical cord is used as an access cannulation site in babies, even after cord clamping. In other words, the blood collected is surely of neonatal origin at this time-point.
The following sentence has been added, Lines: 96 – 97: Currently, not many publications are available concerning UCBGA in dogs, most probably due to different umbilical cord morphology in dogs than in humans and significantly lower birth weight and thus limited vessel size in puppies.
Line 86: In order to determine if the acidosis is respiratory, you should determine the pCO2. Please, clarify.
The sentence has been completed as follows:
The most important measurements are the pH, partial pressure of carbon dioxide (pCO2) and base deficit (BE), which help to determine whether the acidosis is metabolic or respiratory
Material and methods:
You should place the full description of the study population in the MM section. Some very important information should be provided:
The MM section was improved and corrected to meet all the Reviewers suggestions. The changes were as follows:
How many puppies in total and how many puppies per bitch?
The following information was added in the Results section (Lines 204 – 205): Eighty-five puppies were born in 18 Cesarean section. There were delivered 1 – 6 puppies in each Cesarean section.
- It is important to mention how the vet estimated the date for performing the c-section. Were all pregnancies full-term?
The following information was added, Lines 114 - 118: The Cesarean section was scheduled based on progesterone measurements during heat and LH peak determination and was performed on 63 - 64 day after LH peak, depending on the clinical evaluation of the dam, presence of milk and US fetal monitoring (mainly HR).
- Please describe in detail the neonatal management: did the authors subject puppies to any stimulation procedure after birth?
The full section was added as follows Lines 155 - 175:
Depending on the size of the litter 2 – 3 staff members with similar experience and skills were given the first aid and if necessary performed resuscitation. Immediately after placental detachment each puppy was placed on a heated blanket, the heart rate (HR), respiration rate (RR), mucous membrane colour, muscle tone and reflex irritability were briskly evaluated. Then, depending on the initial state, neonatal care or resuscitation was initiated. Neonatal care started from removing of free fluid from the upper airways using the syringe bulb and clamping of the umbilical cord. Thereafter breathing was stimulated by a vigorous rubbing with a towel and the puppy was checked for obvious physical defects. The responding pups i.e. vocalising and breathing spontaneously were moved into a veterinary incubator (Rcom MX-B90N). The non-improving pups underwent neonatal resuscitation. If the heart beat was present, the oxygen (constant flow or mask) was supplied and a gentle stimulation of abdominal area was started. The GV26 acupuncture point was stimulated and vigorous massage was continued. In puppies with asystole cardiac compression was performed at a rate 120 per minute and epinephrine was administered in the jugular vein. If there were no improvement after 20 min of intensive care it was stopped and the puppy was considered dead. All puppies were evaluated at 5th and 20th min using Apgar scoring and then a full clinical examination was performed around moment of discharge to make sure all neonates can be reunited with the dam and sent home. Each owner was contacted by phone 48 h and 7 days after Cesarean section to confirm survival/death of the puppies.
- Did the authors perform a neonatal resuscitation maneuver before blood collection?
The following sentences were re-written, Lines 139 – 143:
Directly after the puppy extraction from the uterus the fetal membranes were torn and removed from the muzzle and the umbilical cord was double clamped before placenta detachment. First forceps was placed as close to the placenta as possible, and the second forceps about 1 cm away from newborn's abdominal wall.
- Please, describe in detail how you performed the blood collection: by venous puncture (which needle caliper?) or by manual ejection.
The following information was added. Lines 144 - 146:
The vessels were gently stretched between forceps to allow the blood sampling. At least 100 mcl of mixed umbilical cord blood was collected into a heparinized syringe with 25G needle and taken for further analysis.
- Surprisingly, authors have managed to collect umbilical blood after cord clamping, as the vessels walls collapse down immediately after blood flow interruption. Please, explicit all the difficulties in collecting umbilical blood.
Before started the research authors practiced and improved the sampling procedure. We were able to collect umbilical cord blood from such small breed like Pomeranian or Yorkshire Terrier. Unfortunately, despite our efforts, not all attempts were successful. We were able to collect cord blood in about half of the investigated puppies (85 born and only 42 sampled). The main difficulty was the amount of blood collected. It was necessary to obtain at least 100 mcl of blood to perform the analysis. Authors observed that it was not directly related to the size of the dam, some vessels collapsed immediately after clamping and some remained turgid and allowed for easy collection regardless the dam or puppy size. The authors suppose it was rather related to the individual umbilical cord conformation and the circulating blood tension. However, this is just observation and needs further investigations, which authors hope to be able to study in future.
- What was the volume of blood collected?
The following information was added Line 145: at least 100 mcl
- Another important information is what vessel direction the umbilical blood was collected? The umbilical cord has a two-way direction of blood flow, i.e., from maternal or fetal origin. Please clarify if the blood was actually from the fetus / neonate.
We clamped the umbilical cord starting form the maternal side and then on the puppy side, allowing for the blood accumulation of the puppy’s origin. The blood for analysis was then collected from the most full vessel. The authors hope such approach minimised the risk of the collection of placental/maternal origin blood.
- Please, detail what procedures you performed for neonatal resuscitation: manual ventilation, mechanical ventilation, exogenous oxygen, etc...
The full section addressing the neonatal care and resuscitation was added: Lines 155 -175
Depending on the size of the litter 2 – 3 staff members with similar experience and skills were given the first aid and if necessary performed resuscitation. Immediately after placental detachment each puppy was placed on a heated blanket, the heart rate (HR), respiration rate (RR), mucous membrane colour, muscle tone and reflex irritability were briskly evaluated. Then, depending on the initial state, neonatal care or resuscitation was initiated. Neonatal care began with removing of free fluid from the upper airways using the syringe bulb and clamping of the umbilical cord. Thereafter breathing was stimulated by a vigorous rubbing with a towel and the puppy was checked for obvious physical defects. The responding pups i.e. vocalizing and breathing spontaneously were moved into a veterinary incubator (Rcom MX-B90N). The non-improving pups underwent neonatal resuscitation. If the heart beat was present, the oxygen (constant flow or mask) was supplied and a gentle stimulation of abdominal area was started. The GV26 acupuncture point was stimulated and vigorous massage was continued. In puppies with asystole cardiac compression was performed at a rate 120 per minute and epinephrine (0,01 mg/kg starting dose and continued if necessary at 0.1 mg/kg) was administered in the jugular vein. If there were no improvement after 20 min of intensive care it was stopped and the puppy was considered dead. All puppies were evaluated at 5th and 20th min using Apgar scoring and then a full clinical examination was performed around moment of discharge to make sure all neonates can be reunited with the dam and sent home. Each owner was contacted by phone 48 h and 7 days after Cesarean section to confirm survival/death of the puppies.
Line 110: I wonder how was possible to collect blood from the umbilical cord of a 3.5kg bitch.
As the authors mentioned above we did not observe huge differences with collection of cord blood in bigger or smaller puppies. We practiced and improved the sampling procedure, before starting the study, and we were able to collect umbilical cord blood from such small breed like Pomeranian or Yorkshire Terrier. The quick collapse of cord vessels happened regardless the size of the dam. For this moment authors cannot provide any explanation for this. We reckon there could be a relationship with individual umbilical cord conformation or blood pressure. However, the latter one is less likely, as we observed differences within the same litter - in some pups sampling was unsuccessful while in the rest it was feasible.
Line 149-156: Was the blood acid-base analysis normalized by the neonate body temperature?
Yes, the temperature correction (37°C) was applied. As the blood was collected immediately after puppies' removal from the uterus, the authors assumed that the newborns’ temperature to be around 37°C.
Line 168: please describe in detail the neonatal intensive care.
As mentioned above full section addressing the neonatal care and resuscitation was added: Lines 155 -175
Line 182: revise your statistical method. The T Test is commonly used to determine if the means of two sets of data are significantly different from each other, which is not the case. You have three sets of data (or groups). Please, clarify.
The authors fully agree with the Reviewer and want to explain that the analysis of variance has been performed but we did not clearly stated it in the MS. We are very sorry for overlooking this and the following description has been added in the Statistical analysis paragraph:
One way repeated measures analysis of variance was used to compare Apgar scores within different time points. The results UCBGA were compared among groups using one way analysis of variance. In case where the third group was too small, t-test was used to compare two sets of data.
Results:
Line 198-199: Have you examined puppies for any mal-formation or necroscopically examined post-mortem?
Puppies were checked for obvious malformations during initial neonatal care, and this information was added to the section on Neonatal care and resuscitation, Lines: 163 – 164. The post-mortem examination was not performed, but this is a very interesting point and the authors wold try to include this in future.
Lines 231-232: I do not think that comparing between the blood acid-base imbalance at birth and further AS (5 min and 20 min) is meaningful. It is better to perform a logistic regression analysis, calculating the odds ratio of each blood gas variable for predicting the first hour clinical outcome of the neonates.
Authors fully agree with the reviewer, we attempted to perform the logistic regression analysis, but due to the low number of cases in some of the investigated groups, it did not return convincing results. The authors hope that with more samples in future we will be able to use logistic regression analysis for comparison of the AS and umbilical cord blood parameters.
Discussion
Line 260: On the other hand, we cannot predict future outcome, such as linear growth development, future behavioral changes, and social interrelationship. In other words, vitality score performed immediately after birth cannot predict long-term outcome. However, it may prove fruitful to predict short-term abnormalities.
The following sentence was added in the Discussion section, Lines 319 - 321: Considering the above, the vitality scoring performed immediately after birth seem to be useful in predicting short-term abnormalities, but it does not provide information on the long-term outcome, and the way of puppy handling might play a role in the subsequent neonatal performance.
Lines 261-262: Indeed, your vitality scores at birth are comparatively lower compared to previous studies (please, refer to Pereira et al., J Vet Med Sci 2020; 82(2):247-253), with special regards to the high frequency of severe distressed puppies.
The following sentence has been added, Lines 328 – 331: . The exact time of cord clamping could also influence the neonatal state as reported by Pereira et al. who described a higher vitality in puppies in which the umbilical cord was preserved intact for a least 3 min after birth to allow the residual blood to flow to the puppies [Pereira].
Lines 262-263: If you performed the initial scoring before the immediate medical attention, probably it will be very low, almost resembling the intra-uterine environment. This has to be clearly stated in your methods and additionally discussed here.
The MM section was improved, and the sentence in Discussion was re-written as follows (Lines: 318 – 322): Our Apgar score results at 0 minute were much lower than those reported by Vilar et al. [22], however it might be due to the different surgical and anesthetic protocols used or the exact time of the initial Apgar assessment, which in our study was performed prior to any medical attention provided for a newborn.
Lines 275-276: even if you have performed an intensive critical care? Please clarify if the percentage of death includes the puppies in which you applied the reanimation protocol.
The sentence has been re-written as follows, Lines 336 - 338:
It is worth noticing that 11.9% (n = 5) of pups did not survived the first 24 hours, did not survived the first 24 hours, despite neonatal critical care provided,
Line 268: what was your neonatal care protocol?
According to the Reviewer suggestions the MM section was improved and the full paragraph was added regarding the neonatal care and resuscitation, Lines 155 – 175.
Line 287: what was your resuscitation protocol?
According to the Reviewer suggestions the MM section was improved and the full paragraph was added regarding the neonatal care and resuscitation, Lines 155 – 175.
Line 288: The Apgar scoring was initially proposed (Virginia Apgar, 1952) to be performed around the first hour of life. Therefore, any clinical assessment of the newborn after 60 min should not be named Apgar score. Please, revise.
Authors are sorry for this mistake, it was corrected as follows, Lines 347 - 348: All newborns underwent a full clinical examination before discharging to ensure all puppies are well enough to go home.
Line 321: this affirmation can only be made with the analysis of blood BE and bicarbonate, which are markers of the metabolic component of acidosis.
We agree that this interpretation is inappropriate. The sentence has been changed.
Interestingly, the 2 puppies which died within first 20 min after delivery had higher initial lactate levels (4.42 and 3.3 320 mmol/L, respectively).
Line 326: important to say that this was vaginally born puppies, whereas your results relate to surgical birth.
The information about the mode of delivery was added, Lines 357 – 358.
Line 337: you cannot discharge an ongoing metabolic acidosis only by analysis lactate concentration. An overall interpretation of BE and bicarbonate should be done.
Additional information has been added to the sentence
However, the puppies examined in our research had a low mean concentrations of lactates (2.45 mmol/L), bicarbonates on lower border (22.78 mmol/L), decreased BE(b) (-6.04 mmol/L) and moderate increased partial pressure of CO2 (58.59 mmHg) which suggest both, respiratory and metabolic component as the reason of the pH changes. This is in agreement with other papers where authors reported the mixed acidosis being the cause of poor neonatal performance.
Lines 353-355: Despite not having an umbilical blood, please see Lucio et al. Domest Anim Endocrinol. 2021;74:106483.
We thank for this reference. The paragraph has been changed as follow:
Since, the glucose concentration decreases with impaired utero-placental perfusion or increased fetal consumption [20], the mode of delivery in our study (elective Caesarian section) vs. eutocia and dystocia resolved via Caesarian section in Vassalo et al. [25] research may explain the presented discrepancies. These findings were also confirmed by Lucio et al. (2021) showing differences in maternal and neonatal glucose level depending on delivery mode. Authors found hyperglycemia in puppies from C-section and fetal dystocia groups whereas infants from eutocia and maternal dystocia remained normoglycemic. One of the proposed theories suggested hyperglycemia is a response to peripartum stress (increasing catecholamine, hypoxia ect.). This explanation may support our findings that newborns with low Apgar scores had higher glucose. On the other hand, Lucio et al. [37] and Vanspranghels et.al [27], suggested that glucose level reflects rather maternal physiology than fetal status and did not appear to be useful for predicting neonatal morbidity.
Conclusion
Line 390-392: this is not a conclusion of your work.
The sentence was re-written as follows:
Based on the presented results it can be stated that the mild acidemia was present in all the investigated puppies regardless the Apgar score result
Lines 392-393: you cannot conclude this.
It was re-written as follows:
The detailed evaluation of a newborn state is crucial in overall understanding of neonatal physiology. In our study the newborn puppies had mild acidemia with elevated pCO2 levels and the HCO3 at the lower range of normal limits suggesting the mixed component in the acidemic state. However the lack of reference values in the available literature make it very difficult to interpret the obtained data.
Reviewer 4 Report
The manuscript ‘animals-1100528’ seems to have been written very quickly and paying little attention to details.
There are many typing mistakes, starting from the Title (Assessment). In the ‘Simple summary’, line 13, the verb is missing; line 15,‘scores’. Abstract, line 31,‘reaching AS’. Introduction, line 95,‘contrast’? Discussion, line 326, ‘academic’; line 342,‘allow’; line 354,‘concentration’; line 363,‘showed’. Conclusion, line 387,‘relation’. Besides, ‘cesarean’ is the usual word, not cesarian. The first sentence of the Introduction is not suitable for a scientific journal (lines 44-45).
In the Results, two pups are missing at the evaluation at 20 minutes (36+2+2) (lines 196-199).
The Discussion is very unclear and the topics covered are not consequential. Some parts, i.e. from line 322 to line 347, should be placed before line 292.
Although the subject is interesting, some questions should have been better addressed. For example, the reason for cesarean sections should have been explained, since the mortality rate (over 10% in the subset of the 42 examined newborns) is rather high for an elective CS.
The choice of measuring the APGAR score immediately after extraction from the uterus, even before neonatal assistance, is rather unusual. Most previous works report that the first measure was taken ‘within five minutes’ from birth.
Some references are unsuitable for a scientific work (13,14); number 6 is a PhD thesis and this should be specified; number 21 needs complete re-writing.
Author Response
We thank the Reviewer for pointing out these mistakes. All of them have been corrected in the manuscript. The whole MS has been checked and typing and spelling errors have been corrected throughout the text. The first sentence of Introduction has been changed to:
The transition from fetus to neonate is a crucial time of physiological adaptation.
In the Results, two pups are missing at the evaluation at 20 minutes (36+2+2) (lines 196-199).
We have added the following info about the size of the "weak" group at 20 min:
At 20 minute of birth, the vast majority of newborns (85.7%, n=36) were scored at 7 or higher. 4 infants (9.5%) were classified as weak (4-6 AS) however, critical pups were also found (4.8.1%, n=2).
The Discussion is very unclear and the topics covered are not consequential. Some parts, i.e. from line 322 to line 347, should be placed before line 292.
The paragraphs have been changed according to the Reviewer's suggestion. The Discussion section have been re-written and corrected.
Although the subject is interesting, some questions should have been better addressed. For example, the reason for cesarean sections should have been explained, since the mortality rate (over 10% in the subset of the 42 examined newborns) is rather high for an elective CS.
The caesarean section in each dam was decided based on a clinical indications such as: previous dystocia history due to uterine inertia, belonging to a breed at high risk of dystocia, a single pup pregnancy, narrow birth canal, older age or other health issues not related to reproduction. Patients in our Clinic which undergo Caesarean section are usually very valuable pedigree dogs, and their offspring are also pure pedigree puppies with high breeding value. As such purebred puppies usually come from a limited gene pool, such litters are at a greater risk of different health related problems and the mortality rate can be slightly higher when compared to overall dogs’ population. However, some authors reported comparable mortality rate in newborn puppies: 9% of deaths within the first 24 hours (Fusi et al. 2020), or up to 7.5% in French bulldogs within first 12 hours of life (Vilar et al. 2018).
Fusi, J., Faustini, M., Bolis, B. et al. Apgar score or birthweight in Chihuahua dogs born by elective Caesarean section: which is the best predictor of the survival at 24 h after birth?. Acta Vet Scand, 2020, 62, 39.
Vilar, J.M.; Batista, M.; Pérez, R.; Zagorskaia, A.; Jouanisson, E.; Díaz-Bertrana, L.; Rosales, S. Comparison of 449 3 Anesthetic Protocols for the Elective Cesarean-Section in the Dog: Effects on the Bitch and the Newborn Pup- 450 pies. Anim Reprod Sci, 2018, 190, 53–62.
The choice of measuring the APGAR score immediately after extraction from the uterus, even before neonatal assistance, is rather unusual. Most previous works report that the first measure was taken ‘within five minutes’ from birth.
Authors fully agree with the Reviewer that Apgar is usually measured a couple of minutes after birth. But Authors’ main interest in this study was to assess puppies at the transition time form fetal to neonatal life, thus it was decided to do the initial Apgar scoring before any assistance. Authors hoped it would better correlate with umbilical cord blood parameters would be more relevant for describing the transition moment.
Some references are unsuitable for a scientific work (13,14); number 6 is a PhD thesis and this should be specified; number 21 needs complete re-writing.
Both references 13 and 14 (Higgins, C. Umbilical-Cord Blood Gas Analysis. Article from: acutecaretesting. org, 2014; Guideline, N.I.C. NCT Policy Briefing: NICE Intrapartum Care Guideline Care of Healthy Women and Their Babies during Childbirth) have been removed.
We have added the info that reference 6 is a PhD thesis.
Reference 21 has been re-written correctly.
Round 2
Reviewer 3 Report
The manuscript has improved considerably. The authors should be commended for their efforts to improve the manuscript. Authors have made changes in the Materials and Methods section to provide more information on the experimental design. More importantly, authors have mentioned and thoroughly discussed the important scientific flaws of the present research. Additionally, substantial modifications have been performed throughout.
Further comments are as follow:
Abstract
Lines 37-41: please, improve your English writing and grammar.
Materials and Methods
Line 168: why did you have to stretch the vessels? I think that is really a matter of the blood vessel diameter and not the length. Please, clarify.
Lines 249-254: please, state that blood gas analysis was normalized by puppies' body temperature, estimated as 37oC.
Results
Lines 287-288: how many puppies did you really use in each litter (per dam)?
Discussion
Lines 653-655: The sentence seems award. Please, revise.
Line 678: “seems”
Author Response
We thank the Reviewer for a constructive and comprehensive review and for the appreciation of our work. We highly value the Reviewer’s comments which really help to improve our manuscript.
Abstract
Lines 37-41: please, improve your English writing and grammar.
The sentences were changed to:
Overall, the puppies with higher glucose levels had lower Apgar scores and were at higher risk of death. Furthermore, in our study the newborn puppies had mild acidemia with elevated pCO2 levels and the HCO3 at the lower range of normal limits suggesting the mixed component in the acidemic state.
Materials and Methods
Line 168: why did you have to stretch the vessels? I think that is really a matter of the blood vessel diameter and not the length. Please, clarify.
The authors meant that the vessel was held between clamps that way to allow the needle to pass through the wall of the blood vessel. As sometimes we found the vessel’s wall bending under the needle and not allowing for its puncture.
To avoid any doubt the word ‘stretch’ was removed and the sentence was re-arranged as follows:
The clamped vessel was then positioned between clamps to allow blood collection.
Lines 249-254: please, state that blood gas analysis was normalized by puppies' body temperature, estimated as 37oC.
The sentence was added:
The blood gas analysis was normalized by puppies' body temperature, estimated as 37°C.
Results
Lines 287-288: how many puppies did you really use in each litter (per dam)?
The information has been added.
The collection of umbilical cord blood was successful in 42 of them (1 to 5 puppies per dam), and these pups were enrolled in the study, in the remaining 43 the amount of the collected blood was not enough to perform analysis.
Discussion
Lines 653-655: The sentence seems award. Please, revise.
The sentence has been changed to:
In human medicine both umbilical cord blood and Apgar score were proved to provide valuable information on neonatal status.
Line 678: “seems”
corrected
Reviewer 4 Report
The work has been extensively modified. Writing errors remain, eg. line 298: we cannot define a canine newborn an 'infant'.
Author Response
We thank the Reviewer for a constructive and comprehensive review and for the appreciation of our work.
Writing errors have been corrected throughout the whole manuscript.
At 20 minute of birth, the vast majority of newborns (85.7%, n=36) were scored at 7 or higher, 4 newborns (9.5%) were classified as weak (4-6 AS) however, critical pups were also found (4.8%, n=2).